# Gene Co-Expression Analysis Reveals Functional Differences Between Early- and Late-Onset Alzheimer’s Disease

**DOI:** 10.3390/cimb47030200

**Published:** 2025-03-18

**Authors:** Abel Isaías Gutiérrez Cruz, Guillermo de Anda-Jáuregui, Enrique Hernández-Lemus

**Affiliations:** 1Computational Genomics Division, National Institute of Genomic Medicine, Mexico City 14610, Mexico; isaiascruz2311@gmail.com; 2Center for Complexity Sciences, Universidad Nacional Autónoma de México, Mexico City 04510, Mexico; 3Investigadores por Mexico, Consejo Nacional de Ciencia y Tecnología (CONAHCYT), Mexico City 03940, Mexico

**Keywords:** Alzheimer’s disease, molecular signatures, gene co-expression networks, pathway enrichment, network analytics

## Abstract

The rising prevalence of Alzheimer’s disease (AD), particularly among older adults, has driven increased research into its underlying mechanisms and risk factors. Aging, genetic susceptibility, and cardiovascular health are recognized contributors to AD, but how the age of onset affects disease progression remains underexplored. This study investigates the role of early- versus late-onset Alzheimer’s disease (EOAD and LOAD, respectively) in shaping the trajectory of cognitive decline. Leveraging data from the Religious Orders Study and Memory and Aging Project (ROSMAP), two cohorts were established: individuals with early-onset AD and those with late-onset AD. Comprehensive analyses, including differential gene expression profiling, pathway enrichment, and gene co-expression network construction, were conducted to identify distinct molecular signatures associated with each cohort. Network modularity learning algorithms were used to discern the inner structure of co-expression networks and their related functional features. Computed network descriptors provided deeper insights into the influence of age at onset on the biological progression of AD.

## 1. Introduction

Alzheimer’s disease (AD) stands as one of the most pressing health challenges of the 21st century, with its prevalence escalating in tandem with global aging demographics [1,2]. AD presents a significant and escalating global health challenge, with its prevalence and associated costs expected to rise substantially in the coming decades. As of 2020, more than 55 million people worldwide were living with dementia, and this number is projected to nearly double every 20 years, reaching approximately 78 million in 2030 and 139 million by 2050. In the United States alone, approximately 6.9 million Americans aged 65 and older are currently living with Alzheimer’s dementia, a figure expected to increase to 13.8 million by 2060 unless medical breakthroughs alter the trajectory of the disease. These projections highlight the growing burden of AD and the pressing need for improved diagnostic and therapeutic strategies (https://www.alzint.org/about/dementia-facts-figures/dementia-statistics, accessed on 2 February 2025) [3].

The economic impact of AD is also substantial. In 2022, the estimated healthcare costs associated with AD treatment in the United States reached USD 321 billion, a figure expected to exceed USD 1 trillion by 2050 due to the increasing prevalence of the disease and the growing demand for long-term care. The financial burden extends beyond healthcare expenses, as families and caregivers bear significant out-of-pocket costs related to home care, assisted living, and lost productivity. On an individual level, the lifetime cost of care for a person with Alzheimer’s disease was estimated at approximately USD 360,000 in 2019, more than double that of individuals without the disease. These statistics underscore the urgent need for effective interventions and support systems to address the rising burden of Alzheimer’s disease on individuals, healthcare systems, and societies worldwide (https://www.ajmc.com/view/the-economic-and-societal-burden-of-alzheimer-disease-managed-care-considerations, accessed on 2 February 2025, https://www.managedhealthcareexecutive.com/view/the-cost-burden-of-alzheimer-s, accessed on 2 February 2025).

These data emphasize the growing need for a multifaceted approach to managing Alzheimer’s disease, including advancements in early detection, improved care strategies, and the development of disease-modifying therapies. With rising cases and costs, investing in research, healthcare infrastructure, and caregiver support is essential to reducing the global burden of Alzheimer’s disease. However, despite extensive research, AD’s etiology and progression remain poorly understood, necessitating innovative approaches [4,5]. In recent years, computational systems biology and network science have emerged as powerful tools for dissecting the intricate molecular mechanisms underlying complex diseases like AD [6,7,8]. Leveraging high-throughput data and advanced analytical methods, these interdisciplinary fields offer new avenues for understanding the pathogenesis of AD and identifying potential therapeutic targets [9,10,11,12,13,14].

Alzheimer’s disease (AD) is a progressive neurodegenerative disorder that primarily affects cognitive function, leading to memory loss, impaired reasoning, and ultimately the inability to perform daily activities. It is the most common cause of dementia, accounting for approximately 60–80% of all dementia cases. The disease is characterized by the accumulation of abnormal protein aggregates in the brain, including extracellular amyloid-beta (Aβ) plaques and intracellular tau tangles, which disrupt neuronal communication and trigger neuroinflammation and cell death. The progressive neuronal degeneration leads to brain atrophy, particularly in regions associated with memory and executive function, such as the hippocampus and cerebral cortex. While the exact etiology of Alzheimer’s disease remains incompletely understood, it is believed to result from a complex interplay of genetic, environmental, and lifestyle factors. The apolipoprotein E (APOE) ϵ4 allele is the strongest known genetic risk factor for late-onset AD, while aging, cardiovascular health, and metabolic dysfunction also contribute to disease onset and progression.

Clinically, Alzheimer’s disease progresses through several stages, beginning with mild cognitive impairment (MCI), where affected individuals experience subtle memory lapses that do not yet interfere significantly with daily life. As the disease advances to moderate and severe stages, symptoms become more pronounced, including disorientation, language difficulties, personality changes, and ultimately, a loss of independent function. In the final stages, patients require full-time care as they lose the ability to communicate, recognize loved ones, and control basic bodily functions. Currently, there is no cure for Alzheimer’s disease, and available treatments primarily focus on symptom management. Pharmacological interventions, such as cholinesterase inhibitors (donepezil, rivastigmine, galantamine) and N-methyl-D-aspartate (NMDA) receptor antagonists (memantine), provide modest cognitive benefits but do not halt disease progression. Recent advances in monoclonal antibody therapies targeting amyloid-beta, such as lecanemab and aducanumab, have shown potential in slowing cognitive decline, but their clinical benefits remain under investigation. Given the increasing prevalence and devastating impact of Alzheimer’s disease, ongoing research efforts aim to uncover novel therapeutic targets, improve early diagnosis, and develop interventions that address the underlying mechanisms of neurodegeneration.

Numerous risk factors have been implicated in the development of AD, including genetic predisposition, aging, and cardiovascular conditions [15,16,17,18,19]. However, the interplay between these factors and their impact on disease progression remains poorly understood [20,21,22]. Addressing this knowledge gap requires a holistic approach that integrates diverse data types and employs sophisticated analytical frameworks. Here, we present a comprehensive investigation into Alzheimer’s disease progression, emphasizing the interaction between genetic predisposition, aging, and cardiovascular conditions. Utilizing data from the Religious Orders Study and Memory and Aging Project (ROSMAP), this study aims to identify molecular networks and pathways that advance understanding of the interplay between these risk factors [8].

By delineating gene expression patterns, elucidating enriched biological pathways, and constructing gene co-expression networks, we aim to unravel the molecular underpinnings of AD pathogenesis. Through the lens of computational systems biology and network science, we seek to uncover novel insights into the complex role of gene expression as it relates to AD onset. This study integrates traditional biomedical research with advanced computational methodologies, providing a novel perspective on AD and paving the way for personalized therapeutic strategies.

## 2. Methods

A flow chart diagram depicting the complete workflow for this project is presented in Figure 1. All the associated code is available in a public repository (see https://github.com/IsaiasGutierrezCruz/BiologiaDeRedesAplicadaAEnfermedadesDelSNC, accessed on 2 February 2025). In the following subsections, we further elaborate on each of these steps and their associated methods and parameters.

### 2.1. Data Acquisition, Preprocessing, and Differential Expression

For our study, we utilized RNA sequencing (RNA-seq) data from the Religious Orders Study and Memory and Aging Project (ROSMAP) [8,23]. Bulk RNA-seq data from samples of the dorsolateral prefrontal lobe were used. A flowchart showing data selection is presented in Appendix A.

ROSMAP provides a robust longitudinal dataset, offering detailed molecular and clinical insights into aging and Alzheimer’s disease progression. Its extensive neuropathological data make it well suited for studying disease mechanisms. In this sense, for our analysis, the data were broken into two groups: early-onset Alzheimer’s disease, with patients diagnosed before age 70, and late-onset Alzheimer’s disease for patients diagnosed at age 70 or older.

The preprocessing of this data involved standard steps to minimize biases typically introduced by batch effects, variations in sequencing depth, and potential contamination. This preprocessing was critical to ensure the integrity and comparability of our expression profiles across different samples [24,25].

Normalization of the data was conducted using the Trimmed Mean of M values (TMM) method [26], implemented via the edgeR package [27]. Following normalization, we conducted differential expression analysis to distinguish between early-onset and late-onset Alzheimer’s disease phenotypes using the same edgeR version 4 tool. This analysis helped identify genes that exhibit significant changes in expression between these two conditions, highlighting their differences at the gene expression level.

### 2.2. Pathway Enrichment Analysis

The functional enrichment analysis was performed using the Generally Applicable Gene-set Enrichment (GAGE) method [28]. Unlike traditional methods that assume uniform expression changes within a pathway, GAGE acknowledges the complexity of biological regulation, where different components of a pathway can be regulated in opposing directions. This nuanced approach allows for a more realistic interpretation of the pathway data.

For these analyses, pathway descriptions were sourced from the Kyoto Encyclopedia of Genes and Genomes (KEGG) [29]. KEGG’s comprehensive databases of signaling and metabolic pathways provided us with the necessary framework to map our expression data onto biologically relevant pathways, therefore providing a thorough description of the molecular mechanisms that are differently affected in early- and late-onset Alzheimer’s disease and identifying key processes impacted by the disease.

### 2.3. Reconstruction of Gene Co-Expression Networks

To explore the complex molecular interplay in Alzheimer’s disease, we inferred gene co-expression networks for both early-onset and late-onset phenotypes using the mutual information metric, implemented in the minet package version 3.20 [30]. Mutual information, a nonlinear measure of association, is particularly adept at capturing the diverse ways in which gene expressions can be interlinked, beyond simple linear correlations. This approach allowed us to construct networks that reflect true biological relationships with higher fidelity. By setting a threshold at the 99th percentile for mutual information values, we ensured that only the most significant connections were considered, thus focusing on the most influential gene interactions. These networks were further analyzed and visualized using the *igraph* package version 2.1.4 [31] and Cytoscape software version 3.10 [32], providing both quantitative and qualitative insights into the gene regulatory networks that define each Alzheimer’s phenotype.

### 2.4. Inference of Modular Structure and Functional Analysis in Alzheimer’s Disease Co-Expression Networks

Modules were detected using the Infomap algorithm [33], as implemented in the igraph package version 2.1.4. We ran 1000 iterations to ensure convergence. We chose the Infomap algorithm for its efficiency and superior performance as demonstrated in various tests, where it consistently ranked highest in runtime, accuracy, and overall performance, particularly in terms of the LFR benchmark [34].

We then proceeded to identify the functional role of these modules using an enrichment strategy. We employed Over-Representation Analysis, utilizing a hypergeometric test to ascertain significant associations between each module’s gene set and sets of genes involved in biological functions, as outlined by the Gene Ontology (GO) database [35]. This approach was implemented using the HTSanalyzer package version 2.3.5 in R version 4.4.1 [36], where GO terms were deemed significantly enriched if they exhibited an adjusted Benjamini–Hochberg *p*-value less than 0.05.

The resultant relationships from the enrichment analysis were represented as functional bipartite networks. These networks consist of two layers: one of gene modules and another of GO terms, effectively illustrating the connections between gene clusters and their related biological processes. This network representation aids in modeling the complex interplay of molecular functions in the progression of Alzheimer’s disease, showcasing the specific biological processes associated with different gene modules.

## 3. Results

Our findings reveal interactions between key Alzheimer’s disease risk factors. For instance, APOE alleles [37,38] show age-dependent effects on disease progression, while cardiovascular biomarkers interact with genetic predisposition to influence molecular network dysregulation.

### 3.1. Differential Gene Expression and Pathway Enrichment Analysis

Differential Gene Expression Analysis (DGE) data (see Figure 2 for a Volcano plot representation of differentially expressed genes in the context of their statistically significant differences and fold change values) were subjected to Pathway Enrichment Analysis (PEA), revealing two significantly perturbed pathways distinguishing the control and study groups. First, the oxidative phosphorylation pathway exhibited negative regulation of complex III and ATPase type V, indicating potential deregulation in metabolite transport and ATP production [39]. Interestingly, the gene encoding ATPase type V displayed minimal interaction with other genes, suggesting a localized alteration within this pathway. Consistent with previous studies, regions affected metabolically displayed reduced neuronal density and increased neurofibrillary tangles [40].

Additionally, the pathway associated with cardiac muscle contraction exhibited significant dysregulation, particularly in genes involved in calcium ion regulation within cardiac muscle cells. Perturbations in these genes may lead to impaired systolic and diastolic processes, potentially contributing to heart failure. Notably, genes encoding Troponin C, calsequestrin 2, and actin demonstrated intermediate fold-change values and significant alterations, suggesting their potential influence on other biological processes.

The relationship between AD and cardiac pathology remains unclear, but evidence suggests a potential link through β-amyloid protein aggregation in both conditions [41,42]. These findings underscore the multifaceted nature of AD pathogenesis and highlight the interconnectedness between neurodegenerative disorders and cardiovascular health.

### 3.2. Gene Co-Expression Networks

Gene co-expression networks were constructed using RNA-seq data from each group, as depicted in Figure 3. Figure 3A illustrates a network with higher gene density concentrated at specific nodes, contrasting with Figure 3B, where greater dispersion is evident. This disparity is further elucidated by examining the node degree distributions presented in Figure 4. Although both networks exhibit similar degree value trends, the control group network displays a larger proportion of nodes with intermediate and low degree values.

Statistical analysis was conducted to assess differences in discrete distribution using $\chi^{2}$ and Cramer’s V testing, yielding significant results: $\chi^{2}$ = 9248727, *p*-value $<2.2\times 10^{-16}$, and Cramer’s V =0.624. These findings underscore distinct distributions between the control and study groups, suggesting potential regulatory differences in gene expression networks associated with Alzheimer’s disease onset age.

### 3.3. Network Modularity and Functional Enrichment

In our study of Alzheimer’s disease, distinct differences in modular distribution were observed between the early-onset Alzheimer’s disease (EOAD) and late-onset Alzheimer’s disease (LOAD) networks. The EOAD network contained 541 modules, compared with only 263 in the LOAD network. Despite the notable difference in the number of modules, the size distribution of these modules remained consistent across both networks. Upon examining modules with at least one enrichment (Figure 5), it was observed that the most enriched modules were not necessarily the largest ones, indicating that module size does not directly correlate with the level of functional significance. For instance, in the control group network, smaller modules like Module 5, with 1216 genes, interact predominantly with cellular components and show significant interactions with the biological function neurogenesis, suggesting intricate biological roles that are not merely dependent on the number of genes.

When these modules were projected onto new networks, the node degree distributions appeared similar, although the EOAD network exhibited a higher number of nodes at low and intermediate degree values, with the nodes having the highest degrees predominantly found in the control group. This observation underscores a complex interplay of gene regulation that might influence disease progression through varying network structures.

The functional enrichment analysis supports this observation by identifying Gene Ontology (GO) terms associated with each module’s biological functions. The EOAD network had 53 distinct GO terms, while the LOAD network had 40. These functional associations, visualized in bipartite graphs, provided insights into the biological dynamics at play. Both networks showed that similar modules, such as Module 2, were linked to the ’neuron part’ GO term, displaying common functional characteristics that suggest shared biological processes across different stages of Alzheimer’s disease. Conversely, unique functional pathways were also evident; for instance, Module 24 in the LOAD network, and corresponding modules in the EOAD network, were linked to the ’defense response’ GO term, indicating distinct responses in disease progression.

Moreover, the rearrangement of modules like the *neurogenesis* module in the LOAD network suggests a significant reconfiguration of biological functions, which may contribute to the differing progression of Alzheimer’s disease between the two groups. Such differences, especially in how modules interact with cellular components and biological functions, highlight the importance of a nuanced approach to understanding Alzheimer’s disease through network modularity and functional enrichment.

This integrated analysis offers deeper insights into the complex biological mechanisms underlying the different stages of Alzheimer’s disease, underscoring the importance of understanding modular and functional variations in disease research. These findings not only enhance our understanding of Alzheimer’s disease mechanisms but also highlight potential targets for therapeutic intervention, demonstrating the value of network approaches in complex disease research.

## 4. Discussion

While this study focuses on computational analyses, we acknowledge the lack of experimental validation as a limitation. However, our findings provide a robust framework for hypothesis generation and prioritization, paving the way for future experimental studies.

The robustness of our approach stems from multiple factors. First, we utilize data from the ROSMAP collaboration, which is among the most well-curated databases for molecular and clinical information on Alzheimer’s disease and related dementias. It is also one of the largest studies including high-throughput NGS transcriptomics. In the present case, after sequencing and data completeness quality controls, we ended up with 174 samples for late-onset Alzheimer’s disease and 47 early-onset Alzheimer’s disease cases.

Another source of robustness is statistical robustness. Network analytics were performed using the mutual information statistical dependency measure. Mutual information is the optimal (maximum likelihood and maximum entropy) estimator of statistical dependency [43,44,45]. It has even been proved using large deviation theory that robust tests of hypotheses can be derived for mutual information [46], giving rise to a closed analytic expression for *p*-values in terms of mutual information lower bounds. Given our sample sizes, these lower bounds gave rise to *p*-values lower than $1\times 10^{-30}$ for all the links considered in the networks analyzed here (see https://github.com/IsaiasGutierrezCruz/BiologiaDeRedesAplicadaAEnfermedadesDelSNC, accessed on 2 February 2025). Modularity optimization was also performed with Infomap [47,48], one state-of-the-art method [49].

In this work, we complement traditional bioinformatic pathways with network-based systems biology approaches to study the differences at the transcriptional level of early-onset and late-onset Alzheimer’s disease. The application of network science to the study of Alzheimer’s disease offers a powerful framework for unraveling the complex molecular interactions that underpin the disease’s progression. By mapping the relationships and interactions among genes and their products, network approaches provide a holistic view of the cellular processes that are disrupted in Alzheimer’s disease. This systems biology perspective enables the identification of key regulatory nodes and modules that are critical in the onset and development of the disease, which may not be apparent through traditional reductionist approaches.

Several genes analyzed exhibited notable log2 fold-change values in the differential gene expression analysis, showcasing that expression patterns are distinct between early-onset Alzheimer’s disease (EOAD) and late-onset Alzheimer’s disease (LOAD). Specifically, pseudogenes related to ribosomes and certain microRNAs were identified as differentially expressed. The presence of differentially expressed pseudogenes, particularly in the LOAD group, suggests potential regulatory changes that could affect the progression and severity of the syndrome. These changes may involve modifications in gene translation that impact ribosomal stability and protein synthesis—processes fundamental to cellular proliferation and neurogenesis [50]. Furthermore, the presence of differentially expressed pseudogenes that interact with ribosomal components highlights the intricate regulatory mechanisms at play, which could lead to varied clinical manifestations of Alzheimer’s disease depending on the onset stage. This observation underscores the importance of re-evaluating the functional capabilities of pseudogenes within the genomic landscape of neurodegenerative diseases.

In addition, as mentioned in the results section, we noticed that the pathway associated with cardiac muscle contraction exhibited significant dysregulation. In particular, genes involved in calcium ion regulation within cardiac muscle cells largely reconfigured their coexpression patterns. These changes may further impact already impaired systolic and diastolic processes, potentially leading to heart failure. Notably, genes encoding Troponin C, calsequestrin 2, and actin demonstrated relevant expression reprogramming, that may, in turn, enhance their potential influence on other biological processes. Furthermore, negative regulation of complex III and ATPase type V within the oxidative phosphorylation pathway suggests potential issues in metabolite transport and ATP production. The minimal interaction of the ATPase type V gene with other genes suggests a localized alteration within this pathway, as regions affected metabolically exhibited reduced neuronal density and increased neurofibrillary tangles [51]. Moreover, it is well known that malfunctioning in Complex I is associated with neurodegenerative diseases, including Alzheimer’s disease [52]. The involvement of Complexes II and III has also been reported, suggesting a broader impact on disease development [53,54].

Structural differences in the network topology further showcase differences between EOAD and LOAD phenotypes beyond individual gene expression levels. While some macroscale features such as component sizes generally agree, the connectivity of individual genes varies between phenotypes. The differences in node degree distributions between the networks further suggest a reconfiguration of gene interactions, indicating that the biological significance of individual genes may shift considerably depending on the stage of Alzheimer’s disease.

The modular structures of the gene co-expression networks were obtained with the understanding that cellular functionality in biological systems can be fractioned into collections of modules, which are responsible for carrying out tasks with characteristics sufficient to differentiate one from another [55]. In the networks analyzed in this study, the biological entities that significantly enriched the modules were consistent, meaning that the entities each module was enriched with belonged to a specific system or purpose. Regarding the modules belonging to the EOAD and LOAD groups, these are enriched in GO terms related to cellular components of brain tissue and neurogenesis; these are the enriched modules that interact with a greater number of nodes in the modular structures. Furthermore, if the two modules from the control group and the two modules from the experimental group that interact with cellular components in terms of transcripts that compose them are compared, a similarity can be observed between modules 5 (EOAD) and 1 (LOAD) and modules 2 (EOAD) and 2 (LOAD).

However, there is a considerable proportion of transcripts that come from other sources, so different behaviors are expected between each Alzheimer’s variant. One may consider, for instance, the evidence for the role of genes in the epigenetic clock and aging, such as elongase ELOVL2 in AD [56].

Considering the aforementioned factors, if there is any disturbance in these nodes, it is possible that it could have a broad repercussion on more communities with which it is interacting, in addition to the implications on the cellular components with which they were enriched. It is noteworthy that when comparing both bipartite networks, a rearrangement can be observed in how they interact with neurogenesis, which may be an indication of some difference in the progression of the disease in patients affected by the two variants under study.

In the control group network, interactions with cellular responses to metals and metabolic processes can also be seen in module 127. These mechanisms do not enrich any module of the LOAD network. Specifically, the absence of metabolism processes in the experimental group is an attractive idea, as a modification in the enrichment of the modules provides information about some disturbance in how these functions are regulated in both Alzheimer’s variants, supporting the previously mentioned results where an overactivation of ribosomes and oxidative phosphorylation were identified. As can be seen, the construction of networks provides more information about the microenvironment that may have caused the appearance of differentially expressed genes.

A relevant issue of the current study is the critical role of microglia in the neurodegenerative process, particularly in maintaining brain homeostasis and responding to pathological hallmarks such as amyloid-beta and tau accumulation. See, for instance, the comprehensive review by Malvaso and collaborators [57], in which the authors highlight that microglial dysfunction, driven by oxidative stress and mitochondrial abnormalities, can lead to either a dystrophic or over-reactive phenotype, exacerbating neurodegeneration. This aligns with our findings, which suggest that dysregulation of key molecular networks in AD includes alterations in microglial-associated pathways, reinforcing the notion that microglia play a central role in disease progression. Our study extends these observations by utilizing network-based analyses to identify key regulatory hubs within these dysregulated pathways, further elucidating the mechanistic underpinnings of microglial dysfunction in AD.

Another important parallel is the focus on integrating molecular, neuropathological, and transcriptomic data to dissect the complexity of microglial states in AD. The review underscores the heterogeneity of microglial responses, including distinctions between senescent, dystrophic, and reactive microglial phenotypes, while also questioning the limitations of animal models in capturing the full spectrum of human microglial behavior. Our work builds upon these insights by leveraging patient-derived multiomics datasets, allowing us to explore microglial contributions to AD progression in a data-driven manner. Moreover, the systematic review introduces a neuropathological scoring system for microglial activation, which could complement our computational approach by providing a histological validation framework for the network perturbations identified in our study. Taken together, both studies contribute to the growing body of evidence that supports microglial dysfunction as a key driver of AD pathology and advocate for integrative, multimodal approaches to fully characterize its impact on disease progression.

Recent findings from the Alzheimer’s Disease Neuroimaging Initiative (ADNI) [58] align with our study in emphasizing the need for more personalized and predictive approaches to understanding Alzheimer’s disease (AD) progression. The ADNI study leverages explainable machine learning (ML) techniques to predict and differentiate AD progression based on sex, focusing on noninvasive clinical features such as neuropsychological test scores and sociodemographic data. Similarly, our study integrates biological network analyses to explore how different risk factors, including genetic predisposition, aging, and cardiovascular conditions, interact in shaping AD progression. While ADNI’s approach highlights sex-based differences in disease progression and prioritizes clinical applicability through an ML-driven decision support system, our work provides complementary insights by elucidating the underlying molecular networks associated with AD onset and progression.

Both studies underscore the importance of utilizing longitudinal data for early disease detection and prognosis. ADNI employs ML models trained on a large cohort to stratify patients based on their likelihood of developing AD within a specific timeframe, refining diagnostic accuracy by incorporating sex-specific predictors. Our study similarly recognizes the value of longitudinal clinical datasets, such as ROSMAP, to investigate how molecular network perturbations evolve over time. Moreover, ADNI’s findings on the significance of neuropsychological tests and demographic factors in predicting AD progression reinforce the relevance of our approach in identifying key molecular hubs that may serve as biomarkers or therapeutic targets. By integrating these computational methodologies, both studies contribute to a growing body of evidence advocating for data-driven, personalized medicine in Alzheimer’s research.

While at this stage this is a descriptive analysis and does not directly lead to actionable clinical insights, these integrated network approaches generate potential leads that can be further explored through in silico, in vitro, and in vivo testing. In our study, highlighting these differences between different manifestations of AD in terms of its temporal manifestation may be important in the context of precision medicine.

### Scope and Limitations

Although systematic and comprehensive on its own scope, this study and the database on which was built upon [8] considered some assumptions that lead to the following limitations:

We classified early-onset AD as a diagnosis before age 70 and late-onset AD at 70 or older. This threshold was selected based on epidemiological studies indicating biological and clinical distinctions at this age. While some classifications define early-onset AD as occurring before age 65 and late-onset AD as occurring after 65, other studies have used 70 years as a cutoff, based both on epidemiological trends and differences in genetic and clinical features. The rationale for our choice is based on the distribution of age at diagnosis in the ROSMAP cohort, ensuring a meaningful comparison between groups while maintaining statistical power. Nevertheless, we acknowledge that alternative classifications exist.

Additionally, dividing participants into early- and late-onset groups may result in an imbalance in sample sizes, which could affect statistical power and the generalizability of our findings. In spite of this, ROSMAP is described as an ideal dataset for studying the molecular mechanisms of AD, but we recognize the need to specify which molecular data were used in our analyses. The dataset includes a wide range of omics data, including transcriptomic, proteomic, and epigenetic profiles, which allow for a comprehensive exploration of disease-related pathways.

Furthermore, while ROSMAP is a well-established cohort for AD research, in this work, we did not explicitly compare it with other large-scale datasets, such as the Alzheimer’s Disease Neuroimaging Initiative (ADNI) or the UK Biobank. This is because ROSMAP was chosen for its extensive neuropathological, molecular, and longitudinal clinical data, which are essential—and largely lacking in other studies—for our network-based approach.

Another important consideration is the potential overlap with other neurodegenerative disorders. Since aging-related comorbidities, such as vascular dementia and Lewy body dementia, can influence molecular signatures and disease progression, it is critical to confirm that our dataset is specific to AD. We state now that participants with other neurodegenerative conditions were excluded from our analysis to minimize confounding effects. Finally, ROSMAP’s longitudinal nature provides a unique opportunity to analyze disease progression over time.

## 5. Conclusions

This study highlights the distinct biological differences between early-onset Alzheimer’s disease (EOAD) and late-onset Alzheimer’s disease (LOAD), underscoring the critical role that the age of onset plays in disease progression. Our analyses of differential gene expression, pathway enrichment, and gene co-expression networks reveal that EOAD and LOAD exhibit unique molecular profiles, which may influence the speed and nature of cognitive decline. Specifically, early-onset cases tend to show faster disease progression and greater disruption of gene networks associated with neurodegenerative processes. In contrast, late-onset cases appear more influenced by cumulative aging factors, including cardiovascular health and lifestyle, which may moderate the pace of decline.

The findings underscore the importance of tailoring AD therapeutic strategies to address the unique molecular and environmental drivers of EOAD and LOAD. By understanding the distinct pathways involved, targeted interventions can be developed to slow disease progression more effectively in both forms of the disease. Future research should continue to explore these differences and integrate additional factors, such as environmental exposures and comorbidities, to provide a more comprehensive picture of Alzheimer’s disease and its heterogeneous nature.

Hence, this study highlights the molecular distinctions between early- and late-onset Alzheimer’s disease through integrative computational approaches. By leveraging gene co-expression networks, pathway enrichment, and modular analyses, we uncovered key differences in biological processes and regulatory mechanisms that underpin each phenotype. These findings not only advance our understanding of Alzheimer’s disease progression but also underscore the value of precision medicine in developing targeted therapeutic strategies tailored to the unique drivers of EOAD and LOAD. Future research should integrate environmental, genetic, and lifestyle factors to provide a more comprehensive understanding of these complex diseases.

## Figures and Tables

**Figure 1 cimb-47-00200-f001:**
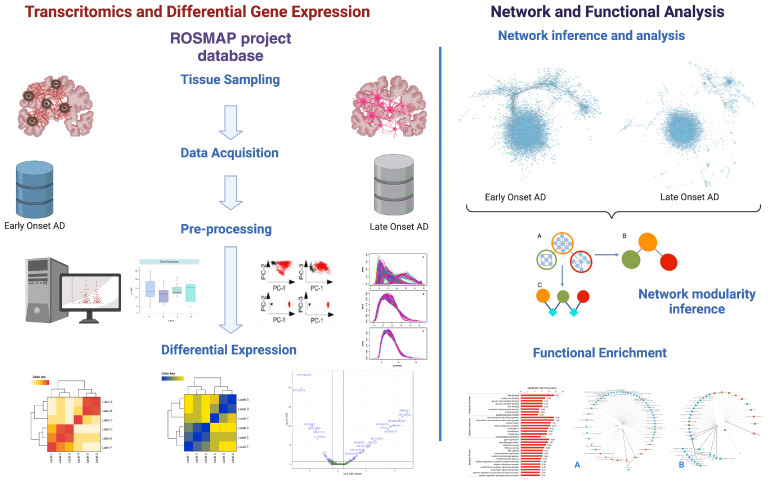
Schematic representation of the workflow followed in this work (created in BioRender.com).

**Figure 2 cimb-47-00200-f002:**
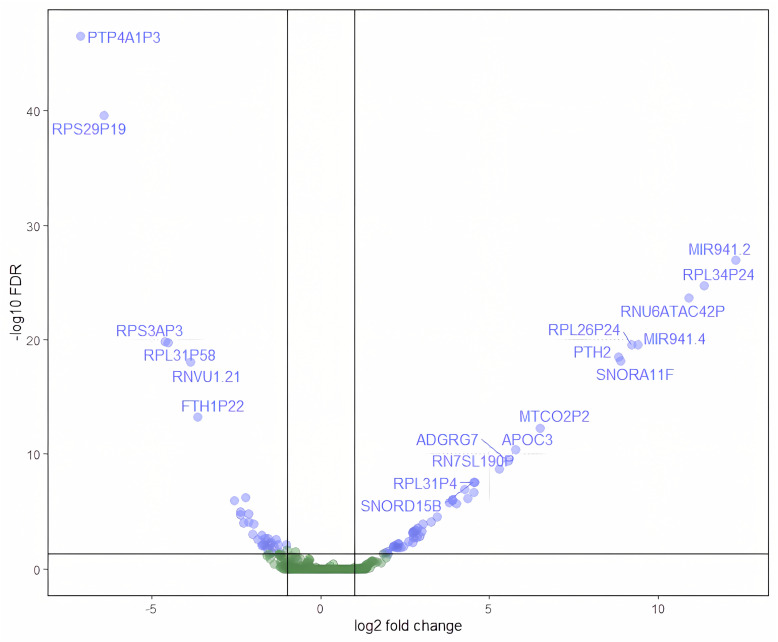
Volcano plot representing differential gene expression between early-onset and late-onset Alzheimer’s disease phenotypes. Each point corresponds to a gene, with the log2 fold change on the x-axis and the negative logarithm (base 10) of the false discovery rate (FDR) on the y-axis. Blue dots correspond to genes with statistical significant expression differences, whereas green genes are not significant. Genes upregulated are shown on the right, and downregulated genes are on the left. Points above the threshold line represent genes with significant changes in expression and potential biological relevance. Gene names are labeled for those with the highest level of significance and fold change.

**Figure 3 cimb-47-00200-f003:**
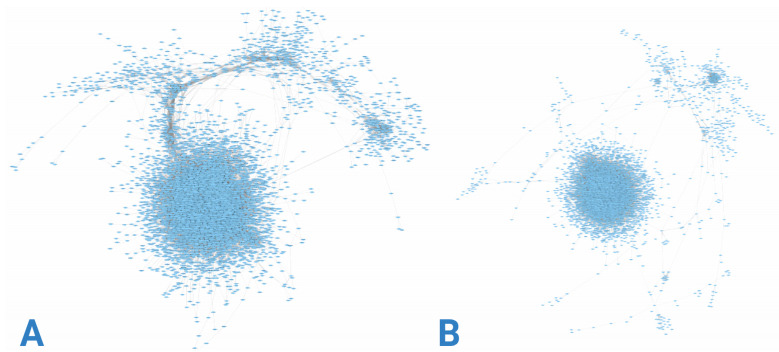
Gene co-expression networks. Nodes represent genes, and edges represent co-expression (measured as significant statistical dependency through mutual information). Panel (**A**) presents the network corresponding to early-onset AD. Panel (**B**) presents the network corresponding to late-onset AD.

**Figure 4 cimb-47-00200-f004:**
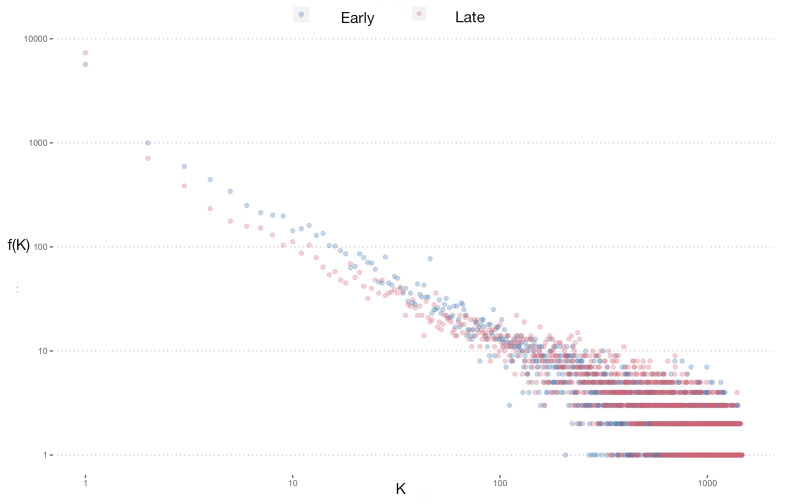
Degree distributions for both networks are illustrated with color-coded nodes: blue dots for early-onset Alzheimer’s disease and red dots for late-onset Alzheimer’s disease. This visualization not only highlights the distinct connectivity patterns unique to each disease stage but also draws attention to the similarities in node density at low and intermediate degrees of connectivity.

**Figure 5 cimb-47-00200-f005:**
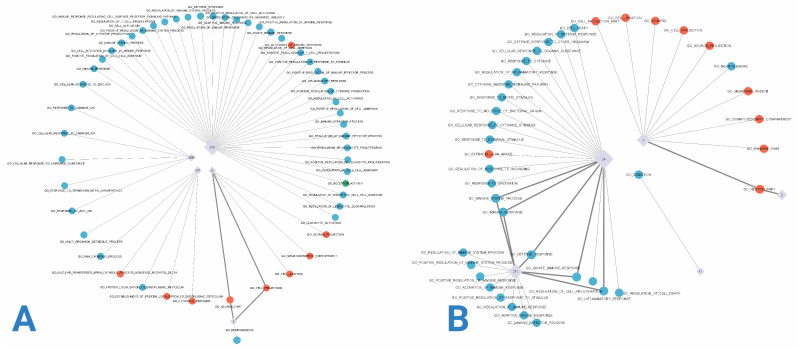
Panels (**A**,**B**) display the bipartite networks for early- and late-onset Alzheimer’s disease, respectively. Each network connects enriched modules, shown as numbered nodes, with their associated Gene Ontology (GO) functions, represented by nodes labeled with GO terms. Edges indicate significant enrichment associations between modules and functions. These networks visualize the unique patterns of functional enrichment linked to the biological processes involved in the early and late stages of Alzheimer’s disease.

## Data Availability

ROSMAP data can be requested at http://www.radc.rush.edu. Accessed on 2 February 2025.

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
