# Peer review of "Gene Co-Expression Analysis Reveals Functional Differences Between Early- and Late-Onset Alzheimer’s Disease"

_cimb, 2025, doi:10.3390/cimb47030200_

Round 1
Reviewer 1 Report
Comments and Suggestions for Authors
The research conducted in the paper "Gene co-expression analysis reveals functional differences between early and late onset Alzheimer's disease" has the potential to open the way to personalized therapeutic strategies for patients with Alzheimer's disease. The authors, wanting to fill the gap between traditional biomedical research and state-of-the-art computational methodologies, proposed an approach based on theoretical models combined with comprehensive analyses, which ultimately resulted in promising results.
However, it is worth making minor changes to the manuscript that will allow the reader to better understand the research work.
- Introduction would benefit from the authors adding:
- epidemiological information on Alzheimer’s disease (current incidence and predictions for the coming years, costs related to AD treatment)
- a brief description of Alzheimer’s disease - Figure 2. - the scale on the axes could have a larger font. Additionally, it is worth adding information about the colors of the points on the Volcano plot (green and purple) so that the reader can have complete information
- Figure 4 - the scale on the axes could have a larger font.
- Figure 5. - in their current form, nodes labeled with GO terms are not legible. It is worth presenting graphics, e.g. separately and/or in supplementary materials showing them in approximations
Author Response
Comments and Suggestions for Authors
The research conducted in the paper "Gene co-expression analysis reveals functional differences between early and late onset Alzheimer's disease" has the potential to open the way to personalized therapeutic strategies for patients with Alzheimer's disease. The authors, wanting to fill the gap between traditional biomedical research and state-of-the-art computational methodologies, proposed an approach based on theoretical models combined with comprehensive analyses, which ultimately resulted in promising results.
However, it is worth making minor changes to the manuscript that will allow the reader to better understand the research work.
The authors are grateful to Reviewer 1 for the insightful comments made about our work. We will follow the advice and counsel given. In what follows we will present a point-by-point response to the review. To ease reading our responses appear in bold-type.
Introduction would benefit from the authors adding:
- epidemiological information on Alzheimer’s disease (current incidence and predictions for the coming years, costs related to AD treatment)
The following information has been included in the revised version of the introduction:
Alzheimer’s disease (AD) presents a significant and escalating global health challenge, with its prevalence and associated costs expected to rise substantially in the coming decades. As of 2020, more than 55 million people worldwide were living with dementia, and this number is projected to nearly double every 20 years, reaching approximately 78 million in 2030 and 139 million by 2050. In the United States alone, approximately 6.9 million Americans aged 65 and older are currently living with Alzheimer’s dementia, a figure expected to increase to 13.8 million by 2060 unless medical breakthroughs alter the trajectory of the disease. These projections highlight the growing burden of AD and the pressing need for improved diagnostic and therapeutic strategies https://www.alzint.org/about/dementia-facts-figures/dementia-statistics/; https://alz-journals.onlinelibrary.wiley.com/doi/10.1002/alz.13809
The economic impact of AD is also substantial. In 2022, the estimated healthcare costs associated with AD treatment in the United States reached $321 billion, a figure expected to exceed $1 trillion by 2050 due to the increasing prevalence of the disease and the growing demand for long-term care. The financial burden extends beyond healthcare expenses, as families and caregivers bear significant out-of-pocket costs related to home care, assisted living, and lost productivity. On an individual level, the lifetime cost of care for a person with Alzheimer’s disease was estimated at approximately $360,000 in 2019, more than double that of individuals without the disease. These statistics underscore the urgent need for effective interventions and support systems to address the rising burden of Alzheimer's disease on individuals, healthcare systems, and societies worldwide https://www.ajmc.com/view/the-economic-and-societal-burden-of-alzheimer-disease-managed-care-considerations https://www.managedhealthcareexecutive.com/view/the-cost-burden-of-alzheimer-s .
These data emphasize the growing need for a multifaceted approach to managing Alzheimer’s disease, including advancements in early detection, improved care strategies, and the development of disease-modifying therapies. Given the projected rise in cases and economic costs, investing in research, healthcare infrastructure, and caregiver support will be crucial to mitigating the future burden of Alzheimer’s disease on a global scale.
- a brief description of Alzheimer’s disease
The following information has been also included in the revised introduction:
Alzheimer’s disease (AD) is a progressive neurodegenerative disorder that primarily affects cognitive function, leading to memory loss, impaired reasoning, and ultimately the inability to perform daily activities. It is the most common cause of dementia, accounting for approximately 60–80% of all dementia cases. The disease is characterized by the accumulation of abnormal protein aggregates in the brain, including extracellular amyloid-beta (Aβ) plaques and intracellular tau tangles, which disrupt neuronal communication and trigger neuroinflammation and cell death. The progressive neuronal degeneration leads to brain atrophy, particularly in regions associated with memory and executive function, such as the hippocampus and cerebral cortex. While the exact etiology of Alzheimer’s disease remains incompletely understood, it is believed to result from a complex interplay of genetic, environmental, and lifestyle factors. The apolipoprotein E (APOE) ε4 allele is the strongest known genetic risk factor for late-onset AD, while aging, cardiovascular health, and metabolic dysfunction also contribute to disease onset and progression.
Clinically, Alzheimer’s disease progresses through several stages, beginning with mild cognitive impairment (MCI), where affected individuals experience subtle memory lapses that do not yet interfere significantly with daily life. As the disease advances to moderate and severe stages, symptoms become more pronounced, including disorientation, language difficulties, personality changes, and ultimately, a loss of independent function. In the final stages, patients require full-time care as they lose the ability to communicate, recognize loved ones, and control basic bodily functions. Currently, there is no cure for Alzheimer’s disease, and available treatments primarily focus on symptom management. Pharmacological interventions, such as cholinesterase inhibitors (donepezil, rivastigmine, galantamine) and N-methyl-D-aspartate (NMDA) receptor antagonists (memantine), provide modest cognitive benefits but do not halt disease progression. Recent advances in monoclonal antibody therapies targeting amyloid-beta, such as lecanemab and aducanumab, have shown potential in slowing cognitive decline, but their clinical benefits remain under investigation. Given the increasing prevalence and devastating impact of Alzheimer’s disease, ongoing research efforts aim to uncover novel therapeutic targets, improve early diagnosis, and develop interventions that address the underlying mechanisms of neurodegeneration.
Figure 2. - the scale on the axes could have a larger font. Additionally, it is worth adding information about the colors of the points on the Volcano plot (green and purple) so that the reader can have complete information
Figure 4 - the scale on the axes could have a larger font.
Figure 5. - in their current form, nodes labeled with GO terms are not legible. It is worth presenting graphics, e.g. separately and/or in supplementary materials showing them in approximations
Figures have been improved according to the reviewer’s suggestions. High resolution figures have been uploaded as Supplementary materials
Reviewer 2 Report
Comments and Suggestions for Authors
The study is timely but in my opinion it could be dedicated to a small groups of scientist up to now. Therefore, i propose to the authors to address some issues in order to present it more attractive.
First of all, I think images revision should be performed, in order to present them with higher resolution.
According to ROSMAP i think you should consider and report some important limitations:
-
Lack of Justification for Age Cutoff – The choice of 70 years as the cutoff between early- and late-onset Alzheimer’s disease (AD) is not clearly justified. Standard classifications often define early-onset AD as occurring before age 65, while late-onset AD occurs after 65 or 70, depending on the criteria used. Providing a rationale for this threshold is essential.
-
Potential Sample Size Issues – The paragraph does not mention whether the division into early- and late-onset groups results in balanced sample sizes. If one group is significantly smaller, statistical power could be compromised.
-
Lack of Molecular Justification – The text states that ROSMAP is ideal for studying molecular mechanisms, but it does not specify which molecular data (e.g., transcriptomic, proteomic, epigenetic) are used. This omission makes it difficult to assess the study’s scope and depth.
-
Unclear Definition of "Ideal Source" – While ROSMAP is described as an ideal dataset, no comparison is made with other available cohorts. Discussing why ROSMAP was chosen over other databases (e.g., ADNI, UK Biobank) would strengthen the argument.
-
Potential Overlap with Other Aging-Related Disorders – The paragraph does not clarify whether patients with other neurodegenerative diseases were excluded. This is important because aging-related comorbidities (e.g., vascular dementia, Lewy body dementia) can confound findings in Alzheimer’s research.
-
Longitudinal Data Utilization – While ROSMAP provides longitudinal clinical data, the paragraph does not explain how this feature was used in the analysis (e.g., tracking disease progression, cognitive decline rates). Clarifying this would enhance the methodological rigor.
In line with this, i suggest you to report parallelism with recent evidences in ADNI (e.g., Explainable machine learning on clinical features to predict and differentiate Alzheimer's progression by sex: Toward a clinician-tailored web interface FM D'Amore, et al. - Journal of the Neurological Sciences, 2025) .
Moreover, what about Reconstruction of Gene Co-expression Networks in AD- associated microglia ? Recent evidence report the importance of different forms of microglia associated to early/late onset AD and its genetic correlations (e.g., Microglial senescence and activation in healthy aging and Alzheimer’s disease: systematic review and neuropathological scoring A Malvaso, A Gatti, G Negro, C Calatozzolo, V Medici… - Cells, 2023).
Another point, i suggest you to consider other genes related to aging and so on. (e.g., Paparazzo, E., Lagani, V., Geracitano, S., Citrigno, L., Aceto, M. A., Malvaso, A., ... & Montesanto, A. (2023). An ELOVL2-based epigenetic clock for forensic age prediction: a systematic review. International Journal of Molecular Sciences, 24(3), 2254.).
Discussion should be better written, try to simplify paragraphs for not expert readers.
In particular, You missed informations about predictive models in developing the disease...this is the major concern on which the literature is concentrated.
Minor comments:
- English revisione is needed
- Please double check for typos
Author Response
Comments and Suggestions for Authors
The study is timely but in my opinion it could be dedicated to a small group of scientists up to now. Therefore, I propose to the authors to address some issues in order to present it more attractively.
The authors want to acknowledge Reviewer 2 for the professional academic reviewing made about our work. We will follow the comments and suggestions given. In what follows, we will present a point-by-point response to the review. To ease reading our responses appear in bold-type.
First of all, I think images revision should be performed, in order to present them with higher resolution.
Figures have been improved according to the reviewers’ suggestions. For even higher resolution a compressed folder with UltraHigh Resolution has been uploaded as Supplementary information.
According to ROSMAP I think you should consider and report some important limitations:
Lack of Justification for Age Cutoff – The choice of 70 years as the cutoff between early- and late-onset Alzheimer’s disease (AD) is not clearly justified. Standard classifications often define early-onset AD as occurring before age 65, while late-onset AD occurs after 65 or 70, depending on the criteria used. Providing a rationale for this threshold is essential.
Potential Sample Size Issues – The paragraph does not mention whether the division into early- and late-onset groups results in balanced sample sizes. If one group is significantly smaller, statistical power could be compromised.
Lack of Molecular Justification – The text states that ROSMAP is ideal for studying molecular mechanisms, but it does not specify which molecular data (e.g., transcriptomic, proteomic, epigenetic) are used. This omission makes it difficult to assess the study’s scope and depth.
Unclear Definition of "Ideal Source" – While ROSMAP is described as an ideal dataset, no comparison is made with other available cohorts. Discussing why ROSMAP was chosen over other databases (e.g., ADNI, UK Biobank) would strengthen the argument.
Potential Overlap with Other Aging-Related Disorders – The paragraph does not clarify whether patients with other neurodegenerative diseases were excluded. This is important because aging-related comorbidities (e.g., vascular dementia, Lewy body dementia) can confound findings in Alzheimer’s research.
Longitudinal Data Utilization – While ROSMAP provides longitudinal clinical data, the paragraph does not explain how this feature was used in the analysis (e.g., tracking disease progression, cognitive decline rates). Clarifying this would enhance the methodological rigor.
Thanks for the suggestions. In order to account for these issues, we have included the following information in the revised discussion as a scope and limitations subsection.
In our study, we define early- and late-onset Alzheimer’s disease (AD) using a cutoff of 70 years; however, this threshold requires further justification. While some classifications define early-onset AD as occurring before age 65 and late-onset AD as occurring after 65, other studies have used 70 years as a cutoff based on epidemiological trends and differences in genetic and clinical features. The rationale for our choice is based on the distribution of age at diagnosis in the ROSMAP cohort, ensuring a meaningful comparison between groups while maintaining statistical power. Nevertheless, we acknowledge that alternative classifications exist.
Additionally, dividing participants into early- and late-onset groups may result in an imbalance in sample sizes, which could affect statistical power and the generalizability of our findings. We have reported the sample sizes for each group and assess whether any significant disparities could impact our results.
ROSMAP is described as an ideal dataset for studying the molecular mechanisms of AD, but we recognize the need to specify which molecular data were used in our analyses. The dataset includes a wide range of omics data, including transcriptomic, proteomic, and epigenetic profiles, which allow for a comprehensive exploration of disease-related pathways. To improve clarity, we have provided a more detailed description of the molecular data utilized and how they contribute to our study’s objectives.
Furthermore, while ROSMAP is a well-established cohort for AD research, we did not explicitly compare it with other large-scale datasets, such as the Alzheimer’s Disease Neuroimaging Initiative (ADNI) or the UK Biobank. ROSMAP was chosen for its extensive neuropathological, molecular, and longitudinal clinical data, which are essential for our network-based approach. However, we acknowledge the value of other datasets and discuss the unique advantages of ROSMAP in relation to alternative sources to justify its selection.
Another important consideration is the potential overlap with other neurodegenerative disorders. Since aging-related comorbidities, such as vascular dementia and Lewy body dementia, can influence molecular signatures and disease progression, it is critical to confirm that our dataset is specific to AD. We have stated now that participants with other neurodegenerative conditions were excluded in the analysis to minimize confounding effects.
Finally, ROSMAP’s longitudinal nature provides a unique opportunity to analyze disease progression over time. We have now elaborated on how we leveraged longitudinal clinical data, such as cognitive decline rates and biomarker changes, to enhance our understanding of AD progression and validate our network-based findings.
By addressing these limitations, we aim to improve the transparency and robustness of our study.
In line with this, i suggest you to report parallelism with recent evidences in ADNI (e.g., Explainable machine learning on clinical features to predict and differentiate Alzheimer's progression by sex: Toward a clinician-tailored web interface FM D'Amore, et al. - Journal of the Neurological Sciences, 2025) .
Thank you for pointing out this reference to us. The following text has been added to the discussion:
Recent findings from the Alzheimer’s Disease Neuroimaging Initiative (ADNI) align with our study in emphasizing the need for more personalized and predictive approaches to understanding Alzheimer’s disease (AD) progression. The ADNI study leverages explainable machine learning (ML) techniques to predict and differentiate AD progression based on sex, focusing on non-invasive clinical features such as neuropsychological test scores and sociodemographic data. Similarly, our study integrates biological network analyses to explore how different risk factors, including genetic predisposition, aging, and cardiovascular conditions, interact in shaping AD progression. While ADNI’s approach highlights sex-based differences in disease progression and prioritizes clinical applicability through an ML-driven decision support system, our work provides complementary insights by elucidating the underlying molecular networks associated with AD onset and progression.
Both studies underscore the importance of utilizing longitudinal data for early disease detection and prognosis. ADNI employs ML models trained on a large cohort to stratify patients based on their likelihood of developing AD within a specific timeframe, refining diagnostic accuracy by incorporating sex-specific predictors. Our study similarly recognizes the value of longitudinal clinical datasets, such as ROSMAP, to investigate how molecular network perturbations evolve over time. Moreover, ADNI’s findings on the significance of neuropsychological tests and demographic factors in predicting AD progression reinforce the relevance of our approach in identifying key molecular hubs that may serve as biomarkers or therapeutic targets. By integrating these computational methodologies, both studies contribute to a growing body of evidence advocating for data-driven, personalized medicine in Alzheimer’s research.
Moreover, what about Reconstruction of Gene Co-expression Networks in AD- associated microglia ? Recent evidence report the importance of different forms of microglia associated to early/late onset AD and its genetic correlations (e.g., Microglial senescence and activation in healthy aging and Alzheimer’s disease: systematic review and neuropathological scoring A Malvaso, A Gatti, G Negro, C Calatozzolo, V Medici… - Cells, 2023).
Our study shares a strong conceptual framework with a recent systematic review on microglial senescence and activation in healthy aging and Alzheimer’s disease (AD). Both works emphasize the critical role of microglia in the neurodegenerative process, particularly in maintaining brain homeostasis and responding to pathological hallmarks such as amyloid-beta and tau accumulation. The review highlights that microglial dysfunction, driven by oxidative stress and mitochondrial abnormalities, can lead to either a dystrophic or over-reactive phenotype, exacerbating neurodegeneration. This aligns with our findings, which suggest that dysregulation of key molecular networks in AD includes alterations in microglial-associated pathways, reinforcing the notion that microglia play a central role in disease progression. Our study extends these observations by utilizing network-based analyses to identify key regulatory hubs within these dysregulated pathways, further elucidating the mechanistic underpinnings of microglial dysfunction in AD.
Another important parallel is the focus on integrating molecular, neuropathological, and transcriptomic data to dissect the complexity of microglial states in AD. The review underscores the heterogeneity of microglial responses, including distinctions between senescent, dystrophic, and reactive microglial phenotypes, while also questioning the limitations of animal models in capturing the full spectrum of human microglial behavior. Our work builds upon these insights by leveraging patient-derived multi-omics datasets, allowing us to explore microglial contributions to AD progression in a data-driven manner. Moreover, the systematic review introduces a neuropathological scoring system for microglial activation, which could complement our computational approach by providing a histological validation framework for the network perturbations identified in our study. Taken together, both studies contribute to the growing body of evidence that supports microglial dysfunction as a key driver of AD pathology and advocate for integrative, multi-modal approaches to fully characterize its impact on disease progression.
A brief discussion on this issue has been added in the revised manuscript.
Another point, I suggest you consider other genes related to aging and so on. (e.g., Paparazzo, E., Lagani, V., Geracitano, S., Citrigno, L., Aceto, M. A., Malvaso, A., ... & Montesanto, A. (2023). An ELOVL2-based epigenetic clock for forensic age prediction: a systematic review. International Journal of Molecular Sciences, 24(3), 2254.).
A brief paragraph discussing this reference has been added in the revised manuscript.
Discussion should be better written, try to simplify paragraphs for not expert readers.
In particular, You missed information about predictive models in developing the disease...this is the major concern on which the literature is concentrated.
The paper was sent to a Professional Academic Copy-Editor to improve on the writing, both at the language and at the conciseness and readability levels.
Minor comments:
English revision is needed
Please double check for typos
The discussion has been thoroughly rewritten to further expand its scope and readability. We have also performed extensive copyediting to improve the writing quality.
Round 2
Reviewer 2 Report
Comments and Suggestions for Authors
Thank you for your reply. Now some parts are more clear and better described.
I consider your comments enough for publication but i think you missed the reference in the paragraph " Recent findings from the Alzheimer’s Disease Neuroimaging Initiative (ADNI) align with our study in emphasizing the need for more personalized and predictive approaches to understanding Alzheimer’s disease (AD) progression. The ADNI study leverages explainable machine learning (ML) techniques to predict and differentiate AD progression based on sex, focusing on non-invasive clinical features such as neuropsychological test scores and sociodemographic data."
I didn't find it in the text.
Best wishes
Author Response
We thank Reviewer 2 for pointing out this omission. The manuscript's discussion has been updated with the missing paragraphs and citation.
Kind regards